# Awareness and Knowledge of HPV, HPV Vaccination, and Cervical Cancer among an Indigenous Caribbean Community

**DOI:** 10.3390/ijerph19095694

**Published:** 2022-05-07

**Authors:** Zachary Claude Warner, Brandon Reid, Priscilla Auguste, Winnie Joseph, Deanna Kepka, Echo Lyn Warner

**Affiliations:** 1Department of Internal Medicine, University of Arizona, 1501 N Campbell Avenue, Tucson, AZ 85724, USA; 2Department of Family Medicine, University of Kansas, 138 N Santa Fe Ave, Salina, KS 67401, USA; breid2@kumc.edu; 3Department of Family Medicine, University of Arkansas for Health Sciences, Little Rock, AR 72205, USA; priscillaauguste@mail.rossmed.edu; 4Salybia Health Clinic, Saint David Parish, Bataka 00109, Dominica; winfran67@hotmail.com; 5College of Nursing, University of Utah, 10 South 2000 East, Salt Lake City, UT 84112, USA; deanna.kepka@hci.utah.edu (D.K.); echo.warner@nurs.utah.edu (E.L.W.); 6Huntsman Cancer Institute, Cancer Control and Population Sciences, 2000 Circle of Hope Drive, Salt Lake City, UT 84112, USA

**Keywords:** women’s health, Caribbean region, cervical cancer, global health, health beliefs, HPV, HPV vaccine, indigenous peoples, knowledge

## Abstract

Caribbean women experience a cervical cancer incidence rate that is three times higher than that among their North American counterparts. In this study, we performed a needs assessment of the knowledge and awareness of HPV, HPV vaccination, and cervical cancer and receipt of cervical cancer screening among an indigenous Caribbean community. We purposively recruited individuals aged ≥18 from a community health care clinic (*n* = 58) to complete a 57-item structured interview including items on demographics, cancer history, knowledge and awareness of HPV, HPV vaccines, cervical cancer, and cervical cancer screening. Participants’ mean age was 47.1 years (SD: 14.4). Most were female (74.1%), were married/partnered (51.7%), had primary education (63.8%), and identified as Kalinago (72.4%). Whereas 79.5% had heard of cervical cancer, few had heard of HPV (19.6%) or the HPV vaccine (21.8%). Among those who knew someone with cancer, 90.9% had heard of the HPV vaccine, compared with only 9.1% of those who did not know anyone with cancer (*p* = 0.02). Access to HPV vaccination is an immediate, cost-effective cancer prevention priority for reducing the disproportionate burden of HPV-related cancers, particularly cervical cancer, in the Caribbean. We recommend culturally targeted education interventions to improve knowledge about HPV vaccination and the link between HPV and cervical cancer.

## 1. Introduction

Cervical cancer is the fourth-leading cause of cancer-related mortality among women worldwide and one of the most common malignancies among women in lower-income countries [1]. Given its high disease burden, preventing cervical cancer is a high priority for cancer control programs around the globe [2]. Although cervical cancer is a preventable disease, it contributes to more lost years of life than breast, stomach, or lung cancer; tuberculosis; and acquired immune deficiency syndrome (AIDS) in Latin America and the Caribbean (LAC) [3,4]. Based on population growth and projected burden, the cervical cancer incidence in LAC is expected to climb 19–132% by 2025 [5]. Papanicolaou (Pap) tests and human papillomavirus (HPV) vaccination have reduced the incidence and mortality of cervical cancer in higher-income countries by up to 80% [6]. However, due to the difficulty of implementing Pap screening programs and lack of access to the HPV vaccine, the burden of cervical cancer remains disproportionately high in low- and middle-income countries (LMICs), with more than 85% of worldwide cervical cancer deaths occurring in these regions [6].

The burden of cervical cancer in LIMCs is not evenly distributed, with some of the highest incidence occurring in indigenous communities. In general, cancer is the second-leading cause of death among indigenous people, and their survival is lower than that among non-indigenous individuals [7]. This high vulnerability to poor cancer-related outcomes is perpetuated by severe health disparities in indigenous communities. For example, many of the factors that contribute to poorer cancer health outcomes in general are prevalent in indigenous communities, such as late stage at diagnosis and limited access to cancer treatment [8]. Furthermore, the strong association between poverty and cancer likely impacts indigenous communities, who are among the poorest in LAC [9,10].

About 10% of the Caribbean population identifies as an indigenous race or ethnicity [7]. However, very little is known about and almost no research has evaluated this population’s knowledge and awareness of cervical cancer prevention and screening. In 2008, Munoz and colleagues recommended primary (prophylactic HPV vaccination) and secondary (introduction and improvement of screening programs) approaches as immediately needed prevention strategies for reducing the burden of cervical cancer in LAC [3]. Research documenting HPV vaccine acceptability and cultural influences is a critical need in this region [3] because limited awareness of cancer prevention and screening opportunities may exacerbate the current disparities in cancer-related morbidity and mortality among indigenous Caribbean populations. This includes the suboptimal uptake of cancer prevention measures [11], which may lead to delays in the diagnosis and treatment of cervical lesions [12].

To the best of our knowledge, no studies have evaluated knowledge and awareness of HPV, HPV vaccination, or cervical cancer prevention and screening among indigenous Caribbean communities. Therefore, in this study, we developed a community partnership to assess the knowledge and awareness of preventable cancers among an Indigenous community in Dominica, West Indies. Evaluating disparities by social grouping, such as identification with an indigenous community, is a common approach to studying inequalities in health [13]. Informed by a constructionist–emancipatory philosophical approach, we aimed to facilitate an opportunity for community members to share their opinions and have a voice in developing priorities for community-based cancer prevention education. A community-based participatory research (CBPR) collaboration [14] was formed between a community public health nurse, a nursing aide, tribal leadership, medical students, and public health researchers to identify cancer-related knowledge and awareness priorities.

## 2. Materials and Methods

### 2.1. Theoretical Framework

The health belief model (HBM) provided a theoretical framework for this study. The HBM is a health behavior framework, first conceptualized by the United States Public Health Service in the 1950s, that has been widely used in public health research to describe barriers to the acceptance of health prevention and preventive screenings among adults. The HBM incorporates individual perceptions, modifying factors, and the likelihood of action to explain barriers to preventive health behaviors (Figure 1). The HBM was used as a guide for the implementation of this study, the selection of study questions, and the dissemination of study results. This project assessed individual perceptions, modifying factors, and the likelihood of action.

### 2.2. Study Design

This study was a community-based participatory needs assessment of cancer prevention education. Three community partners collaborated on the development, design, and implementation of this study. The partnership consisted of local health care staff from a community clinic, including a nurse and a nursing assistant, a student-led nonprofit group consisting of first- and second-year medical students, and university research faculty and staff. The community partners met multiple times prior to the study to plan the research design, the recruitment, and the study materials (i.e., consent form, survey). Power calculations were not conducted because the goal of this study was to establish a community partnership, and research design and sampling were based on the direction of the Kalinago tribal leadership. Nonetheless, small-sample research is highly valuable despite its limited generalizability [15]. This community-engaged research was grounded in trusting relationships; prioritized the community-identified health topics of importance; and incorporated indigenous ways of knowing and tribal customs in the design, data collection, analysis, and dissemination of results.

### 2.3. Participant Recruitment and Data Collection

Purposive sampling occurred through a community health care clinic from 2015 to 2016 at bimonthly clinic visits. The clinic serves a population of approximately 3000 community members, the majority of whom are descendants and/or members of an indigenous tribe. A cross-sectional survey (see Appendix A) was administered to eligible participants from May to December 2016. Eligible participants included all adults ages 18 and older who attended the community health care clinic and could speak and read English. All individuals who attended the clinic during data collection times were approached and screened for eligibility. The study was explained to eligible participants, who were offered the opportunity to participate while they waited to be seen by a health care provider. Participants who were promptly seen by a provider were able to finish the survey before leaving the clinic. The research team emphasized that respondents’ participation or declining participation in no way influenced their or their loved one’s receipt of health services at the present or in the future. Participants were enrolled after they completed the informed consent process with a trained member of the research team. Per recommendation from the community public health nurse, no compensation was provided for participation in the survey. However, participants were invited to join a health education fair at a later date, where they were provided a meal to show appreciation for their time and invited to provide feedback.

The surveys were completed using pen and paper by trained members of the research team based on oral responses from the participants. The survey responses were then manually entered into Excel by two members of the research team. Double data entry occurred for all surveys to validate data quality.

### 2.4. Sociodemographic and Cancer History Variables

Sociodemographic variables included sex (female, male), age (18–39 years, 40–49 years, ≥50 years), relationship status (married/partnered, single/divorced/widowed), ethnicity (Kalinago, other), education (primary school or less, high school or more), household income (<$5000 Eastern Caribbean dollars (ECD)/<$1872 USD, ≥$5000 ECD/≥$1872 USD representing 25% of the average household income in Dominica), health insurance status (insured, uninsured), birth country (Dominica, other), and language (English only, multilingual (i.e., French Creole)). Cancer history variables included having a personal history of cancer or an immediate family member with cancer or knowing anyone with cancer (yes, no).

### 2.5. Cervical Cancer, HPV, and HPV Vaccine Awareness and Knowledge Outcomes

Three categories of outcomes were evaluated: cervical cancer awareness, HPV awareness and knowledge, and HPV vaccine awareness and knowledge. Cervical cancer awareness was assessed only among women with the following three questions: Have you heard of cervical cancer? (yes, no, don’t know). Do you know what a Pap smear is? (yes, no, don’t know). What do you think the likelihood is of you getting cervical cancer? (very likely—not likely at all). HPV awareness and knowledge were assessed in all participants with the following questions: Have you heard of human papillomavirus or HPV? HPV is not the same as HIV (yes, no, don’t know). HPV is able to cause cervical cancer (true, false, don’t know). Most people have HPV at some point in their lives (true, false, don’t know). HPV vaccine awareness and knowledge were assessed for all participants with the following questions: Before today, have you heard of the HPV vaccine (also known as the cervical cancer vaccine or Gardasil)? (yes, no, don’t know). The HPV vaccine has 1 dose (true, false, don’t know). All awareness questions were dichotomized as yes vs. no/don’t know, and all knowledge questions were dichotomized as correct vs. incorrect/don’t know. There were some missing responses, and these are noted in the tables.

### 2.6. Statistical Analysis

Descriptive statistics were calculated for sociodemographic and cancer history factors. Chi square and Fisher Exact (for cells < *n* = 5) tests were performed to evaluate sociodemographic and cancer history correlates with cervical cancer, HPV, and HPV vaccination awareness and knowledge outcomes. Descriptive and inferential statistics were calculated in Stata 13.1 (StataCorp LLC, College Station, TA, USA, www.stata.com, accessed on 22 May 2017).

## 3. Results

### 3.1. Sociodemographics and Cancer History

The mean age of the 58 participants was 47.1 years (SD: 14.4). In Table 1, it can be seen that most participants were female (74.1%), were married/partnered (51.7%), had completed primary education (63.8%), earned less than $5000 ECD annually (53.4%), and identified with Kalinago ethnicity (72.4%). Although most participants had not been diagnosed with cancer themselves (3.4%), 50% had a family member with cancer and 55.2% knew someone with cancer.

### 3.2. Sociodemographic Correlates of Awareness of Cervical Cancer

Whereas 79.5% of female participants had heard of cervical cancer, nearly all had heard of a Pap smear (92%, Table 2). All women who had not heard of a Pap smear were aged 40–49 years, which is also the most frequent age for being diagnosed with cervical cancer [16]. Participants felt relatively vulnerable to developing cervical cancer, with 61.5% believing they were somewhat to very likely to get cervical cancer in their lifetime.

### 3.3. Sociodemographic Correlates of Awareness of HPV and the HPV Vaccine

Among all participants, few had heard of HPV (19.6%) or the HPV vaccine (23%, Table 3). A higher proportion of those who knew of the HPV vaccine knew someone with cancer (90.1%) compared with those who had not heard of the HPV vaccine (50.0%, *p* = 0.02). There was limited knowledge among participants about the contribution of HPV to cervical cancer (17%) and about the fact that most people get HPV at some point (23%, data not shown).

## 4. Discussion

Cervical cancer is a current public health problem among indigenous peoples in LAC. Despite the fact that the cervical cancer incidence in this region is among the highest in the world, published research in the Caribbean demonstrates a general lack of awareness of cervical cancer, HPV, and the HPV vaccine. In contrast, participants in our study had a relatively high awareness of cervical cancer, and this is likely due to the robust community health education provided at the community clinic from which our sample was drawn. While most participants had heard of cervical cancer, those who had not were in the prime age range for developing cervical cancer, suggesting that targeted efforts may be necessary among women at the ages at which they are most likely to develop cervical cancer. The possibility for reducing cervical cancer health disparities among indigenous communities in LAC is now further underscored by the availability of efficacious HPV vaccines. Our findings emphasize the need for culturally targeted HPV and HPV vaccine educational campaigns, vaccination programs, and government support.

The HPV vaccine was not readily available in this community at the time of the study. Nonetheless, the cost-effectiveness of HPV vaccination far outweighs any other prevention method because treatment for cervical cancer is not readily available or easily accessed in this community; thus, individuals must travel internationally to receive it. Individuals living in this community therefore may face extreme financial hardship in trying to find the means to receive treatment for cervical cancer, even with early detection, which further emphasizes the critical need for HPV vaccine availability. Furthermore, the HPV vaccine also protects against other cancers and HPV-related morbidity. It is available for boys and girls, and the WHO recommends vaccinating both to reduce the heavy burden of HPV. Low knowledge of the HPV vaccine is an issue that needs to be addressed, but the most important next step is making the HPV vaccine readily available. To measure the effectiveness of vaccine programs, there is a need to measure national incidences of cervical cancer and other HPV-related cancers as well as the incidence of common cervical cancer screenings. We recommend future concerted efforts from community and tribal leadership, politicians, and external cancer and vaccine agencies to expand access to the HPV vaccine and implement vaccine registries in Caribbean communities.

The HPV vaccine remains the most cost-effective tool for the prevention of cervical cancer, yet most participants in this study were unfamiliar with HPV and the HPV vaccine. The reason for the limited familiarity with the HPV vaccine in LAC communities, including the rural indigenous community we surveyed, may be limited access to vaccines in general and particularly to the HPV vaccine. Access to HPV vaccination is an immediate, cost-effective cancer prevention priority for reducing the disproportionate burden of HPV and cervical cancer among indigenous LAC communities. Therefore, we recommend a culturally tailored education program that is supported by local and national governing boards to help improve knowledge of HPV vaccination and the link between HPV and cervical cancer. These programs should prioritize HPV vaccination for cervical cancer prevention and may be the most cost-effective method for reducing cervical cancer health disparities in indigenous LAC communities such as the one represented by our findings [3]. Future cost-effectiveness research should consider the impacts of national HPV vaccination programs as alternatives to cervical cancer screening programs.

There are limitations that should be considered when interpreting our findings. First, the cross-sectional nature of this study precludes our ability to assess changes in the knowledge and awareness of cervical cancer, HPV, and the HPV vaccine. However, we were able to show the burden that the lack of knowledge and resources may have on this population, helping to shed light on the need for intervention. Our sample was recruited from the only health care clinic in this community. Although this means that any adult seeking care for any reason would have been identified and offered an opportunity to participate, individuals who did not seek care from the clinic were not included. The intention was to expand the survey recruitment through door-to-door recruitment by research assistants accompanied by trusted community and tribal leaders. However, these recruitment efforts were rendered impossible due to a natural disaster (hurricane), which also delayed data analysis while members of the study team were evacuated from the region. From a demographic perspective, our sample is overwhelmingly female; thus, our results may not accurately represent the knowledge and awareness of males in this community. However, given historical associations of cervical cancer and HPV with the female gender, we expect that knowledge and awareness is likely lower among males than females, further underscoring the need for community-wide culturally tailored information for both males and females. Another limitation worth noting is that our results may not be generalizable. However, as the country we recruited from is not the only LAC community impacted by HPV and cervical cancer, we strongly feel that our results may still help other populations build a foundation to conduct needs assessments to see where the gaps in their countries’ programs may lie. Our small sample size does not have the power to detect statistically significant differences in multivariable regressions; thus, we are limited in our ability to perform more complex analyses to predict associations of sociodemographic and cancer factors with our outcomes. Finally, there are limitations in the use of the HBM to examine knowledge and attitudes regarding cancer and cancer prevention among indigenous communities because it does not take into account the influence of the geographic setting and structural barriers that potentially influence likelihood of action (e.g., the likelihood of engaging in cancer prevention behaviors and screenings) [13]. This theory also does not acknowledge the relative social power and poverty inequalities experienced by this particular indigenous community [13].

## 5. Conclusions

Despite their having some of the highest incidence and mortality rates associated with cervical cancer in the world, there is a distinct gap in the literature on cervical cancer, HPV, and HPV vaccine awareness and knowledge among indigenous LAC communities. This study is a first step toward elucidating the need to improve clinical services and educational opportunities for preventing HPV infection and the development of cervical cancer, among other HPV-related diseases. Improving HPV vaccination (knowledge/awareness/receipt) among indigenous Caribbean communities is an important step toward reaching the UICC and WHO goal of equal access to preventive care to reduce premature cancer deaths by 25%.

## Figures and Tables

**Figure 1 ijerph-19-05694-f001:**
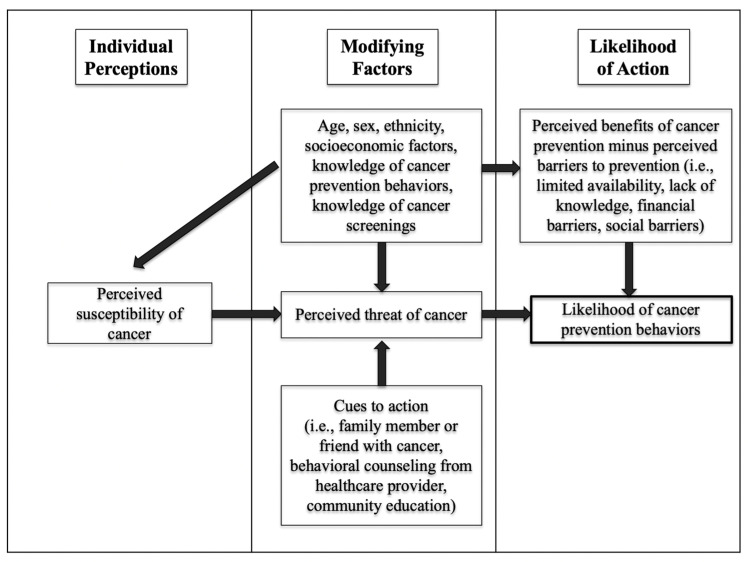
Application of the health belief model.

**Table 1 ijerph-19-05694-t001:** Sociodemographic and cancer history factors (N = 58).

	N	%
**Sex**		
Female	43	74.1
Male	15	25.9
**Age**		
18–39	19	32.8
40–49	12	20.7
≥50	27	46.5
**Relationship status ^1^**		
Married/partnered	30	51.7
Single/divorced/widow	27	46.5
**Ethnicity**		
Kalinago	42	72.4
Other	16	27.6
**Education ^1^**		
Primary school or less	37	63.8
High school or more	20	34.5
**Household Income ^1^**		
<$5000 ECD	31	53.4
≥$5000 ECD	22	37.9
**Health Insurance ^1^**		
Insured	5	8.6
Uninsured	52	89.7
**Birth Country**		
Dominica	56	96.6
Other	2	3.4
**Language ^1^**		
English only	21	36.2
Multilingual	32	55.2
**Personal cancer history**		
Yes	2	3.4
No	56	96.6
**Family cancer history**		
Yes	29	50.0
No	29	50.0
**Know anyone with cancer**		
Yes	32	55.2
No	24	41.4

^1^ Missing for: relationship status *n* = 1, education *n* = 1, income *n* = 5, insurance stats *n* = 1, language *n* = 5.

**Table 2 ijerph-19-05694-t002:** Demographic correlates of cervical cancer awareness among females only (N = 39).

	Heard of Cervical Cancer ^1^		Heard of Pap Smear ^1^		Likelihood of Getting Cervical Cancer ^1^	
	Yes N(%)	No/DK N(%)	Yes N(%)	No/DK N(%)	NL, NLAA ^2^ N(%)	SL, L, VL ^2^ N(%)
**Age**						
18–39	13(41.9)	1(12.5)	14(38.9) **	0(0.0) **	3(30.0)	10(41.7)
40–49	6(19.4)	3(37.5)	6(16.7) **	3(100.0) **	3(30.0)	4(16.7)
≥50	12(38.7)	4(50.0)	16(44.4) **	0(0.0) **	4(40.0)	10(41.7)
**Relationship status**						
Married/partnered	19(61.3)	5(62.5)	22(61.1)	2(66.7)	7(70.0)	14(58.3)
Single/divorced/widowed	12(38.7)	3(37.5)	14(38.9)	1(33.3)	3(30.0)	10(41.7)
**Ethnicity**						
Kalinago	24(77.4)	5(62.5)	26(72.2)	3(100.0)	8(80.0)	17(70.8)
Other	7(22.6)	3(37.5)	10(27.8)	0(0.0)	2(20.0)	7(29.2)
**Education**						
Primary school or less	18(60.0)	7(87.5)	23(65.7)	2(66.7)	8(80.0)	14(60.9)
High school or more	12(40.0)	1(12.5)	12(34.3)	1(33.3)	2(20.0)	9(39.1)
**Household Income**						
<$5000 ECD	14(48.3)	5(83.3)	17(51.5)	2(100.0)	5(62.5)	10(45.5)
≥$5000 ECD	15(51.7)	1(16.7)	16(48.5)	0(0.0)	3(37.5)	12(54.5)
**Health Insurance**						
Insured	4(12.9)	1(12.5)	4(11.1)	1(33.3)	1(10.0)	4(16.7)
Uninsured	27(87.1)	7(87.5)	32(88.9)	2(66.7)	9(90.0)	20(83.3)
**Birth Country**						
Dominica	30(96.8)	8(100.0)	35(97.2)	3(100.0)	10(100.0)	23(95.8)
Other	1(3.2)	0(0.0)	1(2.8)	0(0.0)	0(0.0)	1(4.2)
**Language**						
English only	14(48.3)	2(28.6)	15(44.1)	1(50.0)	11(52.4)	4(40.0)
Multilingual	15(51.7)	5(71.4)	19(55.9)	1(50.0)	10(47.6)	6(60.0)
**Personal cancer history**						
Yes	2(6.5)	0(0.0)	2(5.6)	0(0.0)	1(10.0)	1(4.2)
No	29(93.5)	8(100.0)	24(94.4)	3(100.0)	9(90.0)	23(95.8)
**Family cancer history**						
Yes	17(54.8)	4(50.0)	21(58.3)	0(0.0)	7(70.0)	11(45.8)
No	14(45.2)	4(50.0)	15(41.7)	3(100.0)	3(30.0)	13(54.2)
**Know anyone with cancer**						
Yes	18(62.1)	4(50.0)	20(58.8)	2(66.7)	7(70.0)	14(63.6)
No	11(37.9)	4(50.0)	14(41.2)	1(33.3)	3(30.0)	8(36.4)

** *p* = 0.01; ^1^ Cervical cancer questions missing for *n* = 4 women, ^2^ VL = very likely, L = likely, S = somewhat likely, NL = not likely, NLAA = not likely at all.

**Table 3 ijerph-19-05694-t003:** Demographic correlates of awareness of HPV and the HPV vaccine (N = 58).

	Heard of HPV ^1^		Heard of HPV Vaccine ^2^	
	Yes N(%)	No/DK N(%)	Yes N(%)	No/DK N(%)
**Sex**				
Female	10(90.9)	31(68.9)	10(83.3)	31(72.1)
Male	1 (9.1)	14(31.1)	2(16.7)	12(27.9)
**Age**				
18–39	6(54.5)	13(28.9)	5(41.7)	14(32.5)
40–49	2(18.2)	9(20.0)	3(25.0)	7(16.3)
≥50	3(27.3)	23(51.1)	4(33.3)	22(51.2)
**Relationship status**				
Married/partnered	7(63.6)	23(51.1)	7(58.3)	22(51.2)
Single/divorced/widowed	4(36.4)	22(48.9)	5(41.7)	21(48.8)
**Ethnicity**				
Kalinago	6(54.6)	6(54.6)	9(75.0)	31(72.1)
Other	5(45.4)	5(45.4)	3(25.0)	12(27.9)
**Education**				
Primary school or less	5(45.4)	31(70.4)	8(66.7)	27(64.3)
High school or more	6(54.6)	13(29.6)	4(33.3)	15(35.7)
**Household Income**				
<$5000 ECD	3(42.9)	26(59.1)	6(75.0)	22(52.4)
≥$5000 ECD	4(57.1)	18(40.9)	2(25.0)	20(47.6)
**Health Insurance**				
Insured	1(9.1)	4(8.9)	1(8.3)	4(9.3)
Uninsured	10(90.9)	41(91.1)	11(91.7)	39(90.7)
**Birth Country**				
Dominica	10(90.9)	44(97.8)	12(100.0)	41(95.4)
Other	1(9.1)	1(2.2)	0(0.0)	2(4.6)
**Language**				
English only	6(54.6)	14(34.1)	7(63.6)	13(32.5)
Multilingual	5(45.4)	27(65.9)	4(36.4)	27(67.5)
**Personal cancer history**				
Yes	0(0.0)	2(4.4)	0(0.0)	2(5.6)
No	11(100.0)	43(95.6)	12(100.0)	41(95.4)
**Family cancer history**				
Yes	7(63.6)	22(48.9)	7(58.3)	22(51.2)
No	4(36.4)	23(51.1)	5(41.7)	21(48.8)
**Know anyone with cancer**				
Yes	8(72.7)	24(55.8)	10(90.9) *	21(50.0) *
No	3(27.3)	19(44.2)	1(9.1) *	21(50.0) *

* *p* < 0.05; ^1^ heard of HPV missing for *n* = 2; ^2^ heard of HPV vaccine missing for *n* = 3.

## Data Availability

The datasets used and/or analyzed as part of the present study are available from the corresponding author on reasonable request.

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
