# Peer review of "Awareness and Knowledge of HPV, HPV Vaccination, and Cervical Cancer among an Indigenous Caribbean Community"

_ijerph, 2022, doi:10.3390/ijerph19095694_

Round 1
Reviewer 1 Report
Please find comments attached

Author Response
Reviewer 1
- The scope of this study is really limited considering the sample size. However, the authors have done the best of a limited situation. Future studies should aim at a larger sample size since no statistical analyses can be performed on their current group of respondents. Total 58 patients is about 2% of the estimated 3000 members of the community and is really not representative of even that community
- We understand the reviewer’s concerns about sample size and generalizability. In response, we point to the emergent value of small sample research for studying health disparities, as described in a commentary by Etz and Arroyo, 2015. The authors note that studies which include small sample sizes often are focused on the hardest to reach participants, as is the case in our sample of the remote Indigenous Kalinago tribe in Dominica. Not only is this community difficult to reach physically the culture and community norms require forging trusting relationships between the research team, ethical boards, and Kalinago Tribal leaders to conduct human subject’s research. The premise of this commentary is that small sample research is highly valuable, despite it limits to generalizability. Community engaged research, particularly in Indigenous communities, must be grounded in trusting relationships, appropriately prioritize health issues that are of important to the community, and value Indigenous ways of knowing and tribal customs in the design, data collection, analysis, and dissemination of results. We have added some of this language to the methods section 2.2 Study Design on lines 102-109:
“Power calculations were not conducted because the goal of this study was to establish a community partnership and the research design and sampling were based on the direction of the Kalinago Tribal leadership. Nonetheless, small sample research is highly valuable, despite it limits to generalizability.17 This community engaged research was grounded in trusting relationships, prioritized the community-identified health topics of importance, and incorporated Indigenous ways of knowing and tribal customs in the design, data collection, analysis, and dissemination of results.”
We also now included the following reference to support these statements: “17. Etzl KE, Arroyo JA. Small Sample Research: Considerations Beyond Statistical Power. Prev Sci (2015) 16:1033–1036. doi: 10.1007/s11121-015-0585-4.”
- Why is the monetary threshold of $5000 ECD chosen?
- This threshold was chosen as a proxy for poverty given that it is $5000 ECD is approximately 25% of the average household income in Dominica, WI. We now include this information in the methods section on line 139.
- How long did it take to gather information about 58 subjects? Could the researchers waited say 2-3 times as long to get to 100-150 subjects?
- The research team spent 12 months of bi-monthly trips to the clinic to recruit 58 participants to complete the survey. This initial round of recruitment was limited to the community health clinic based on agreements with tribal leadership, who were wary of outside researchers. Because of the rapport built with the community and tribal leadership, we were invited to expand recruitment in a door-to-door approach. However, this effort was disrupted due to a severe natural disaster, a major hurricane, that devastated the island. The Kalinago territory was the hardest hit region, experiencing severe landslides and loss of life. The research was disrupted at this time because local leadership and clinical partners were focused on lifesaving measures and needs of the community, this rebuilding continues to be the primary focus of the local community to date. Study staff were evacuated and relocated. Nonetheless, given the immense effort required to build community partnerships required to conduct research in indigenous and rural communities the small sample size does not undermine the value of these results. Specific details of this natural disaster can be found online at: https://reliefweb.int/report/dominica/dominica-impact-hurricane-maria-disaster-profile-january-2018. This was originally mentioned in the limitations section of the paper, but we have now revised this statement to be more direct on lines 276-278: “The intention was to expand the survey recruitment through door-to-door recruitment by research assistants accompanied by trusted community and tribal leaders. However, these recruitment efforts were rendered impossible due to a natural disaster (hurricane) which also delayed data analysis while members of the study team were evacuated from the region.”
- On the same line, what is a minimum sample size you would need to perform any statistical analyses?
- Thank you for the opportunity to clarify our statistical analysis. We did perform bivariate statistics and had adequate sample size to do so using appropriate statistical tests including chi-square tests and Fisher Exact tests, which are applied for variables that have cell size smaller than n=5 to account for independence between two variables when comparing groups. Chi-square tests are intended to approximate independence and assume larger samples, in contrast, Fisher Exact tests are exact because approximating is inappropriate with smaller sample sizes (Kim, HY. Statistical notes for clinical researchers: Chi-squared test and Fisher's exact test. Restor Dent Endod. 2017 May; 42(2): 152–155. Published online 2017 Mar 30. doi: 10.5395/rde.2017.42.2.152.).
Reviewer 2 Report
Since there is little research on indigenous Caribbean communities, this study provides valuable information. I suggest this report should add more recent research references, for example, P2L48 "In general, cancer is the second leading cause of death among Indigenous people, and their survival is lower than non-Indigenous individuals." needs a reference. Table 3, the number of male participants is mis-typed.
Author Response
Reviewer 2
- Since there is little research on indigenous Caribbean communities, this study provides valuable information. I suggest this report should add more recent research references, for example, P2L48 "In general, cancer is the second leading cause of death among Indigenous people, and their survival is lower than non-Indigenous individuals." needs a reference.
- Thank you for this recommendation. This passage now has a reference associated (#7), thank you. Much of the literature on this topic is sparsely conducted given that Dominica is among the smallest, regional islands in the Caribbean. We have referenced the most up to date citations that are available, to our knowledge.
- Table 3, the number of male participants is mis-typed.
Thank you for noting this typo. This has been corrected.
Reviewer 3 Report
This cross-sectional study evaluated the awareness and knowledge of HPV, HPV vaccination and cervical cancer among an indigenous Caribbean community. It is an interesting manuscript and the authors correctly point out the inequal burden of cervical cancer between western and low-income countries. I have the following comments:
- In the introduction section the authors should provide the incidences of cervical cancer in Dominica
- Furthermore, in the introduction section the authors should describe what type of screening against cervical cancer is used in Dominica?
- What is the percentage of women vaccinated against HPV and older than 18 in Dominica? If the vaccine was not available in their community were they vaccinated elsewhere?
- Would it be possible to decrease the incidence of cervical cancer in Dominica with HPV vaccination and without implemented screening programme?
- There are only 58 participants in this study. Did the authors perform a power analysis prior to initiation of the study?
- The participants were recruited from May-December 2016. The authors should explain the time gap between data collection and final analysis
- The discussion should be shortened
- How would the authors improve the HPV vaccination rate among indigenous populations? In addition, how would they propose to improve it in some developed countries?
Author Response
Reviewer 3
- In the introduction section the authors should provide the incidences of cervical cancer in Dominica
- We agree with the reviewer that including the incidences of cervical cancer in Dominica would be helpful, but unfortunately there is not published data or a cancer registry in the country from which to reference this data. The most recent values from the Pan American Health Organization for this region are from 2014 and these were previously cited, but are not specific to Dominica. We have included a statement about this in the discussion: “To measure the effectiveness of vaccine programs there is a need to measure national incidences of cervical cancer and other HPV-related cancers as well as common cervical cancer screenings.”
- Furthermore, in the introduction section the authors should describe what type of screening against cervical cancer is used in Dominica?
- There are no national screening programs in Dominica, however our clinical partners discussed occasionally using Papanicolau tests for cervical cancer. We have added a statement about this in the discussion: “To measure the effectiveness of vaccine programs there is a need to measure national incidences of cervical cancer and other HPV-related cancers as well as common cervical cancer screenings.”
- What is the percentage of women vaccinated against HPV and older than 18 in Dominica? If the vaccine was not available in their community were they vaccinated elsewhere?
- This information is not tracked on a national level in Dominica. Anecdotally, we expect this number to be extremely low based on conversations with local clinical teams who had never administered an HPV vaccine.
- Would it be possible to decrease the incidence of cervical cancer in Dominica with HPV vaccination and without implemented screening programme?
- Thank you for this interesting question to consider. HPV vaccines have shown strong efficacy for decreasing incidence of cervical cancer and other types of HPV related cancers and diseases. While this particular question is beyond the scope of the current study we have included a statement about this as a potential area for future research in the discussion: “Future cost-effectiveness research should consider the impact of national HPV vaccination programs as an alternative to cervical cancer screening programs.”
- There are only 58 participants in this study. Did the authors perform a power analysis prior to initiation of the study?
- We understand the reviewer’s concerns about sample size and generalizability. In short, a power calculation was not conducted. This was due to the community-based participatory research design of the study. We followed the guidance of community leaders, specifically Kalinago Tribal leadership in designing the study and developing the sampling strategy. Respecting the Kalinago’s preferences for data collection was instrumental for building trust and establishing a relationship to perform this study. Our response to Reviewer 1’s first comment pertaining to sampling provides additional detail about this decision. We have included this additional context in the methods section, 2.2 Study Design on lines 102-109:
“Power calculations were not conducted because the goal of this study was to establish a community partnership and the research design and sampling were based on the direction of the Kalinago Tribal leadership. Nonetheless, small sample research is highly valuable, despite it limits to generalizability.17 This community engaged research was grounded in trusting relationships, prioritized the community-identified health topics of importance, and incorporated Indigenous ways of knowing and tribal customs in the design, data collection, analysis, and dissemination of results.”
- The participants were recruited from May-December 2016. The authors should explain the time gap between data collection and final analysis
- Thank you for this opportunity to clarify. There was a gap in the completion of final analysis due to a major natural disaster in Dominica that required study staff to relocate and undergo evacuation and resettlement. We included the following statement to clarify on lines 273-278: “The intention was to expand the survey recruitment through door-to-door recruitment by research assistants accompanied by trusted community and tribal leaders. However, these recruitment efforts were rendered impossible due to a natural disaster (hurricane) which also delayed data analysis while members of the study team were evacuated from the region.”
- Additionally, this manuscript was previously submitted to another journal prior to the 2020 COVID-19 pandemic. Unfortunately, due to unforeseen circumstances at the previous journal (e.g., change in editorial staff, transition to a new online review portal, and delays in finding peer reviewers) the final outcome of that submission was only recently reached in March of 2022. We recognize that manuscript publication delays during the COVID-19 pandemic have been widespread.
- The discussion should be shortened
Thank you for this suggestion. To adequately respond to reviewer comments this section of the manuscript is only slightly shorter than our original submission. We also reviewed other recently published manuscripts in the journal to ensure it is in line with expectations for length. We would be willing to revisit and again attempt to shorten the discussion section at the editor’s discretion.
- How would the authors improve the HPV vaccination rate among indigenous populations? In addition, how would they propose to improve it in some developed countries?
- Thank you for the invitation to consider possible opportunities for improving HPV vaccination rates among indigenous populations and other developing countries. Based on the findings of our project and the knowledge gained from our community-based collaboration we included the following three specific recommendations to improve HPV vaccination rates among indigenous populations in the Discussion:
- “Our findings emphasize the need for a culturally targeted HPV and HPV vaccine educational campaigns, vaccination programs, and government support. “
- “We recommend future concerted effort from community and tribal leadership, politicians, and external cancer and vaccine agencies to expand access to the HPV vaccine and implement vaccine registries in Caribbean communities.”
- Thank you for the invitation to consider possible opportunities for improving HPV vaccination rates among indigenous populations and other developing countries. Based on the findings of our project and the knowledge gained from our community-based collaboration we included the following three specific recommendations to improve HPV vaccination rates among indigenous populations in the Discussion:
“Therefore, we recommend a culturally tailored education program that is supported by local and national governing boards to help improve knowledge of HPV vaccination and the link between HPV and cervical cancer. These programs should prioritize HPV vaccination for cervical cancer prevention and may be the most cost effective method for reducing cervical cancer health disparities in Indigenous LAC communities like the one represented by our findings [3].
Round 2
Reviewer 3 Report
The manuscript is improved and all questions have been adequately answered. I believe the manuscript is now suitable for publication